# β-Hydroxy-β-methyl Butyrate Regulates the Lipid Metabolism, Mitochondrial Function, and Fat Browning of Adipocytes

**DOI:** 10.3390/nu15112550

**Published:** 2023-05-30

**Authors:** Geyan Duan, Changbing Zheng, Jiayi Yu, Peiwen Zhang, Mengliao Wan, Jie Zheng, Yehui Duan

**Affiliations:** 1CAS Key Laboratory of Agro-Ecological Processes in Subtropical Region, Hunan Provincial Key Laboratory of Animal Nutritional Physiology and Metabolic Process, National Engineering Laboratory for Pollution Control and Waste Utilization in Livestock and Poultry Production, Institute of Subtropical Agriculture, Chinese Academy of Sciences, Changsha 410125, China; duangeyan21@mails.ucas.ac.cn (G.D.); yujiayi22@mails.ucas.ac.cn (J.Y.); 2College of Advanced Agricultural Sciences, University of Chinese Academy of Sciences, Beijing 100049, China; 3College of Animal Science and Technology, Hunan Agricultural University, Changsha 410128, China; chamdpion@stu.hunau.edu.cn (C.Z.); 1228065671@stu.hunau.edu.cn (P.Z.); wmengliao01@stu.hunau.edu.cn (M.W.)

**Keywords:** β-hydroxy-β-methyl butyrate, lipid metabolism, mitochondrial biogenesis, fat browning, insulin resistance

## Abstract

A growing number of in vivo studies demonstrated that β-hydroxy-β-methyl butyrate (HMB) can serve as a lipid-lowering nutrient. Despite this interesting observation, the use of adipocytes as a model for research is yet to be explored. To ascertain the effects of HMB on the lipid metabolism of adipocytes and elucidate the underlying mechanisms, the 3T3-L1 cell line was employed. Firstly, serial doses of HMB were added to 3T3-L1 preadipocytes to evaluate the effects of HMB on cell proliferation. HMB (50 µM) significantly promoted the proliferation of preadipocytes. Next, we investigated whether HMB could attenuate fat accumulation in adipocytes. The results show that HMB treatment (50 µM) reduced the triglyceride (TG) content. Furthermore, HMB was found to inhibit lipid accumulation by suppressing the expression of lipogenic proteins (C/EBPα and PPARγ) and increasing the expression of lipolysis-related proteins (p-AMPK, p-Sirt1, HSL, and UCP3). We also determined the concentrations of several lipid metabolism-related enzymes and fatty acid composition in adipocytes. The HMB-treated cells showed reduced G6PD, LPL, and ATGL concentrations. Moreover, HMB improved the fatty acid composition in adipocytes, manifested by increases in the contents of n6 and n3 PUFAs. The enhancement of the mitochondrial respiratory function of 3T3-L1 adipocytes was confirmed via Seahorse metabolic assay, which showed that HMB treatment elevated basal mitochondrial respiration, ATP production, H^+^ leak, maximal respiration, and non-mitochondrial respiration. In addition, HMB enhanced fat browning of adipocytes, and this effect might be associated with the activation of the PRDM16/PGC-1α/UCP1 pathway. Taken together, HMB-induced changes in the lipid metabolism and mitochondrial function may contribute to preventing fat deposition and improving insulin sensitivity.

## 1. Introduction

Obesity has spread at an alarming rate and poses a great threat to human health. There are conventional approaches that can work against obesity including diets and exercise. However, these two methods require considerable time and effort, and they are hard to maintain. Therefore, other practicable alternatives such as nutritional interventions are being actively developed.

β-hydroxy-β-methyl butyrate (HMB), a metabolite of leucine, has shown the ability to combat obesity and insulin resistance. For instance, rodents that were given access to HMB supplementation had less fat mass than their counterparts [1,2]. In line with these findings, our previous studies on pigs also suggest that HMB prevents fat accumulation by inhibiting lipogenesis and stimulating lipolysis [3,4]. In the context of high-fat feeding, the administration of HMB could prevent obesity and alleviate insulin resistance in mice [5]. These metabolic benefits exhibited by animal models highlight the need to further research the molecular basis. However, much less is known about the regulatory roles and the molecular mechanisms of HMB in the lipid metabolism of the adipocytes.

The mitochondria play a pivotal role in lipid metabolism, not only by providing key intermediates (such as glyceryl 3-phosphate and acetyl-CoA) for lipogenesis [6], but also by generating energy (such as adenosine triphosphate (ATP)) via fatty acid oxidation (FAO) [6,7]. It has been well documented that the mitochondria are the major site of oxygen consumption and are closely associated with the basal metabolic rate [8]. Great attention has been devoted to the relevance of their oxidative activity to metabolic regulation, and there is convincing evidence of a link between mitochondrial respiratory disruption and obesity [9,10]. A study in humans reported that the mitochondrial aerobic capacity of adipocytes is reduced in obese individuals compared with nonobese individuals [11]. In diet-induced or genetic obese mice, decreased mitochondrial oxidative phosphorylation capacity in white adipocytes was observed [12]. This phenomenon was also found in isolated mitochondria from the subcutaneous adipose depots of obese subjects, along with impaired ATP synthesis capacity [13]. In particular, obese mice showed a compromised mitochondrial FAO capacity in adipocytes [14], which was demonstrated to be a contributing factor to obesity [15]. Thus, mitochondrial respiratory function might hold the key to the lipid metabolism of adipocytes and can be used as a target for the development of promising therapeutic strategies to combat obesity.

Apart from mitochondrial respiratory function, fat browning is also reported to be associated with lipid metabolism. The process of ‘fat browning’ refers to the transformation of white adipocytes into beige adipocytes when exposed to cold environments or other activators, which is called ‘beige’ or ‘halogen’ [16,17]. White adipocytes convert excess nutrients into triglycerides (TGs) within lipid droplets, while brown and beige adipocytes are characterized by the smaller size of lipid droplets and the greater number of mitochondria, dissipating energy via uncoupled thermogenesis. Notably, uncoupling protein 1 (UCP1) is a mitochondrial thermogenic effector that is present in brown and beige adipocytes [18]. As such, mitochondrial biogenesis constitutes an essential part of beige adipocyte induction [19]. Moreover, the browning process is regulated by several key transcription factors, such as PR domain-containing 16 (PRDM16) and peroxisome proliferator-activated receptor-gamma coactivator 1 alpha (PGC-1α), which promote mitochondrial biogenesis and enhance energy expenditure [20,21,22,23]. However, little is known about the regulation of fat browning in adipocytes by HMB.

3T3-L1 adipocytes, an *in vitro* model of white adipocytes, are widely used in studies of obesity and fat browning [24,25]. Hence, this study aims to investigate the effects of HMB on lipid metabolism, mitochondrial respiratory function, and fat browning in 3T3-L1 adipocytes, and explores underlying mechanisms.

## 2. Materials and Methods

### 2.1. Reagents

HMB free acid (purity ≥ 95%), α-ketoisocaproate (KIC) (purity ≥ 98%), dexamethasone, insulin, and 3-isobutyl-1-methylxanthine were purchased from Sigma (St. Louis, MO, USA). Triiodothyronine and rosiglitazone were purchased from Sigma-Aldrich (Sigma-Aldrich Trading Co., Ltd., Shanghai, China). Dulbecco’s modified Eagle’s medium (DMEM), fetal bovine serum (FBS), and penicillin-streptomycin (P/S) were purchased from Gibco (Life Technologies, Grand Island, NY, USA).

### 2.2. Cell Culture 

3T3-L1 preadipocytes were grown in DMEM containing 10% FBS and 1% P/S and cultured in an incubator at 37 °C with 5% CO_2_. After preadipocytes reached about 90% confluence and became growth-inhibited for 2 days, differentiation was triggered by adding dexamethasone (1 µM), 3-isobutyl-1-methylxanthine (0.5 mM), and insulin (10 µg/mL). The differentiation medium was renewed every 2 days until the cells fully differentiated. With the addition of 50 nM triiodothyronine and 1 µM rosiglitazone (browning cocktail) to the differentiation medium, the browning of 3T3-L1 adipocytes was induced. Differentiated adipocytes were treated with the serum-free medium for 6 h (starvation treatment) before other treatments.

### 2.3. Cell Proliferation Analysis

3T3-L1 preadipocytes were seeded in a 96-well plate (2000 cells per well). After 80% confluence, the cells were subjected to starvation treatment for 6 h. Then, cells were refed with DMEM containing varying concentrations of HMB (0, 6.25, 12.5, 25, 50, 100 µM) for 24 h. Next, Cell Counting Kit-8 (Dojindo, Kumamoto Prefecture, Japan) was utilized for cell proliferation analysis. Absorbance was measured at 450 nm by an enzyme-linked immunosorbent assay plate reader (800TS, BioTek, Winooski, VT, USA).

### 2.4. Oil Red O Staining

3T3-L1 adipocytes were rinsed twice using phosphate-buffered saline (PBS) and fixed with 4% paraformaldehyde (4 mL) for 1 h. After removing paraformaldehyde, oil red O solution was added, and the cells were stained for 3 h in darkness. Next, PBS was used for washing the cells (3 times). Photographs were obtained using a microscope. Absorbance was measured at 510 nm using a microplate reader (800TS, BioTek, Winooski, VT, USA).

### 2.5. The Determination of TG Levels

The cell suspension of 3T3-L1 adipocytes was centrifuged at 1000 rpm for 10 min at 4 °C. The cells were then washed via PBS and centrifuged. After resuspension, the cells were homogenized via sonication. Cellular TG levels were measured using a commercial TG assay kit (A110-1-1, Nanjing Jiancheng Bioengineering Institute, Nanjing, China) following the manufacturer’s instructions.

### 2.6. Western Blotting Analysis

Cell samples were collected by using RIPA Lysis Buffer (Beyotime Biotechnology, Shanghai, China) containing protease inhibitor cocktail tablets and phosphatase inhibitor cocktail tablets (Sigma-Aldrich Trading Co., Ltd., Shanghai, China). The next procedures were conducted as previously described [26]. The primary antibodies used were the following: CCAAT/enhancer-binding protein alpha (C/EBPα, 18311-1-AP, Proteintech, Chicago, IL, USA, 1:500); peroxisome proliferator-activated receptor gamma (PPARγ, 2435S, CST, Boston, MA, USA, 1:1000); phosphorylated (p)-AMP-activated protein kinase AMPK (p-AMPK, 2535S, CST, Boston, MA, USA, 1:1000); phosphorylated (p)-silent information regulator 2 related enzyme 1 (p-Sirt1, 2314S, CST, Boston, MA, USA, 1:2000); hormone-sensitive lipase (HSL, 4107S, CST, Boston, MA, USA, 1:1000); lipoprotein lipase (LPL, ab137821, Abcam, Cambridge, Cambs, UK, 1:1000); uncoupled protein 3 (UCP3, 10750-1-AP, Proteintech, Chicago, IL, USA, 1:500); PGC-1α (ab54481, Proteintech, Chicago, IL, USA, 1:3000); PRDM16 (ab106410, Abcam, Cambridge, Cambs, UK, 2 µg/mL); UCP1 (14670, CST, Boston, MA, USA, 1:1000); glyceraldehyde-3-phosphate dehydrogenase (GAPDH, 60004-1-Ig, Proteintech, Chicago, IL, USA, 1:7000). The secondary antibodies were HRP goat anti-rabbit IgG (31460, Proteintech, Chicago, IL, USA, 1:5000) and HRP goat anti-mouse IgG (31430, Proteintech, Chicago, IL, USA, 1:5000).

### 2.7. Enzyme-Linked Immunosorbent Assay (ELISA)

The concentrations of glucose-6-phosphate 1-dehydrogenase (G6PD), LPL, and adipose triglyceride lipase (ATGL) in adipocytes were detected using commercial kits (G6PD assay kit/MM-1073M2, LPL assay kit/MM-0407M2, ATGL assay kit/MM-1076M2) on the basis of the manufacturer’s instructions (Jiangsu Meimian Industrial Co., Ltd., Yancheng, China).

### 2.8. Fatty Acid Composition Analysis

3T3-L1 adipocytes were collected to centrifuge tubes. Next, isooctane (4 mL) was added for extraction (12 h) (heat slightly to 45 °C). An amount of 200 μL of potassium hydroxide-methanol solution (0.4 mol/L) was added to the tube and homogenized using a vortex mixer (30 s) for methyl esterification. Subsequently, about 1 g of sodium bisulfate was added for neutralizing potassium hydroxide, followed by vigorous shaking. The supernatant was filtered through a 0.22 μm filter to a sample bottle and then analyzed using a gas chromatograph (Agilent 7890A, Santa Clara, CA, USA).

### 2.9. Seahorse Metabolic Assay

3T3-L1 preadipocytes were seeded in a 24-well culture plate (Seahorse, Billerica, MA, USA) at a density of 1.5 × 10^5^ cells per well. After differentiation, the cells were treated with HMB (50 µM) for 24 h. Afterward, the cells were washed twice and XF Assay medium containing 4.5 g/L glucose, 1 mM sodium pyruvate, and 2 mM glutamine (adjust the pH to 7.4 ± 0.1 using 1 mol/L NaOH) was added. Before the assay, the plate was placed in a 37 °C incubator without CO_2_ for 1 h. Mitochondrial oxygen consumption rate (OCR) measurement was performed using Seahorse Biosciences XF-24 Analyzer (Seahorse, Billerica, MA, USA). After the measurement of basal respiration, oligomycin (4 µM), carbonyl cyanide p-trifluoromethoxyphenylhydrazone (FCCP) (9 µM), and rotenone/antimycin A (5 µM) were added sequentially to measure ATP production, maximal respiration, and non-mitochondrial respiration, respectively.

### 2.10. Statistical Analysis

Data analyses were conducted using SAS 8.2 software (Institute, Inc., Cary, NC, USA) using one-way analysis of variance. The differences among groups were compared using Duncan’s multiple comparisons. For comparisons between the two groups, two-tailed *t*-test was used. All data were presented as mean ± SEM. The results were considered statistically significant at *p* < 0.05.

## 3. Results

### 3.1. HMB Suppressed Lipid Accumulation in 3T3-L1 Adipocytes

As shown in Figure 1A, compared with the control group (0 µM), the 50 µM HMB treatment increased the optical density (OD) value (*p* < 0.05), suggesting a more active state of cell proliferation. By contrast, the 100 µM HMB supplementation had an inhibitory effect (*p* < 0.05). There were no significant differences between the control group and the other groups (*p* > 0.05). In addition, the HMB treatment (50 µM) attenuated lipid accumulation in the adipocytes, as evidenced by reduced lipid droplets (Figure 1B) and TGs (Figure 1C). Thus, 50 µM was chosen for the subsequent experiments.

To further explore the molecular changes in the lipid metabolism, we measured the expression of some related proteins (Figure 2). In comparison with the control group, the 50 µM HMB treatment downregulated the protein expression levels of C/EBPα (*p* < 0.01) and PPARγ (*p* < 0.05), and upregulated the protein expression levels of p-AMPK (*p* < 0.001), p-Sirt1 (*p* < 0.05), HSL (*p* < 0.05), LPL (*p* < 0.01), and UCP3 (*p* < 0.001). These differences might indicate a blunted lipogenic reaction and augmented catabolism of lipids, which caused a reduced lipid accumulation in the 3T3-L1 adipocytes.

### 3.2. HMB Altered the Concentrations of Lipid Metabolism-Related Enzymes and Fatty Acid Composition in 3T3-L1 Adipocytes

After treating the cells with or without HMB for 24 h, the lipid metabolism-related enzyme concentrations and fatty acid composition were measured. As shown in Figure 3, the HMB group had reduced concentrations of G6PD (*p* < 0.001), LPL (*p* < 0.001), and ATGL (*p* < 0.001) compared with the control group.

Table 1 shows the effects of HMB on the fatty acid composition of the 3T3-L1 adipocytes. Compared with the control group, the HMB treatment significantly increased the contents of C10:0 (*p* < 0.05), C18:0 (*p* < 0.01), C20:4n6 (*p* < 0.001), C22:6n3 (*p* < 0.001), ∑n6 PUFA (*p* < 0.001), and ∑n3 PUFA (*p* < 0.001), whereas the contents of C14:0 (*p* < 0.05), C15:0 (*p* < 0.05), C16:1 (*p* < 0.01), and C17:0 (*p* < 0.05) were reduced by HMB administration.

### 3.3. HMB Improved the Mitochondrial Respiratory Function of 3T3-L1 Adipocytes

To assess the effect of HMB on the mitochondrial respiratory function of the 3T3-L1 adipocytes, the Seahorse metabolic assay was conducted. As shown in Figure 4, compared with the control group, the HMB treatment increased the basal mitochondrial respiration by 37% (*p* < 0.05), enhanced ATP production by 27% (*p* < 0.01), and elevated H^+^ leak by 56% (*p* < 0.05). Additionally, the maximal respiration and non-mitochondrial respiration were increased by 20% (*p* < 0.01) and 21% (*p* < 0.05), respectively.

### 3.4. HMB Promoted the Browning of 3T3-L1 Adipocytes

We then looked into the protein expression of fat browning markers to test the hypothesis that HMB may stimulate the browning of adipocytes. As shown in Figure 5, compared with the control group, all three treatments significantly upregulated the protein expression levels of PGC-1α (*p* < 0.05), PRDM16 (*p* < 0.05), and UCP1 (*p* < 0.05). The highest levels of these proteins were observed in the cells treated with the browning cocktail, followed by those incubated with HMB. Compared with the browning cocktail and HMB, KIC showed the minimal effect on promoting the protein expression of PGC-1α, PRDM16, and UCP1.

## 4. Discussion

In the face of overnutrition, adipocytes store TGs and display a hypertrophic phenotype. Our results show that the 50 µM HMB treatment reduced lipid deposition (as manifested by reduced lipid droplets) in the 3T3-L1 adipocytes, confirming the lipid-lowering effect of HMB in vivo [3,5]. To explore the molecular mechanisms behind the suppression of fat accumulation, we further detected the expression of several proteins involved in lipid accumulation (C/EBPα and PPARγ) and lipolysis (AMPK and Sirt1). C/EBPα and PPARγ are central nuclear transcription factors that are responsible for lipid accumulation [27]. We found that the two regulators exhibited diminished protein expression levels after the HMB treatment, accompanied by a decreased cell TG content. In line with these observations, we observed that the concentrations of G6PD, LPL, and ATGL were markedly reduced in response to the HMB treatment. G6PD is a key enzyme required for lipogenesis, and its increase is closely connected to the disturbance of lipid metabolism and insulin resistance in obesity [28,29]. LPL plays an important role in fat deposition and is regarded as a marker of adipocyte differentiation [30]. As for ATGL, it possesses both lipolytic and lipogenic properties [31]. Hence, these findings indicate that HMB prevents lipid accumulation in part through PPARγ and C/EBPα regulation. As a metabolite of leucine, our results confirm and extend the observation made by Sun et al. [32] by demonstrating that leucine led to a marked decrease in the mRNA expression of fatty acid synthase in the 3T3-L1 adipocytes. These findings suggest that HMB may provide benefits similar to leucine in inhibiting lipid accumulation in the 3T3-L1 adipocytes. On the other hand, AMPK oversees the energy status and acts as a master regulator of metabolism [33]. Under the conditions of low cellular ATP levels, AMPK is activated and favors catabolism over anabolism. Similarly, Sirt1 is another important sensor that modulates metabolic processes such as FAO and lipolysis in response to limited nutrient availability [34,35]. In the present study, the HMB treatment (50 µM) significantly increased the expression levels of p-AMPK and p-Sirt1, as well as the expression of HSL (a major enzyme for lipolysis) [36] and UCP3 (an indicator of the intense degree of FAO) [37]. These findings suggest that HMB promoted the lipolysis of the 3T3-L1 adipocytes partially through the AMPK/Sirt1 signaling. These *in vitro* findings are consistent with the observations obtained in pigs, where HMB might spur lipolysis via the AMPK/Sirt1 axis in the adipose tissue [3]. Thus, the activation of the AMPK/Sirt1 signaling might reflect a low energy supply, thereby stimulating catabolic pathways in lipid metabolism. Moreover, Sirt1 can repress PPARγ transcription [35], which may then result in attenuated fat deposition. Therefore, HMB might blunt fat deposition via suppressing lipid synthesis, promoting lipolysis, and enhancing FAO, and these effects might be associated with the AMPK/Sirt1-mediated PPARγ signaling.

Apart from the total lipid content, studies on the effects of HMB treatments on regulating fatty acid composition have gained our attention due to the emerging obesity-linked features being identified for n3 and n6 PUFAs. For example, the contents of n3 and n6 PUFAs in adipose tissue were reported to be negatively correlated with the adipocyte size [38], indicating that these kinds of fatty acids may limit fat accumulation and adipocyte hypertrophy. Further evidence confirmed that genetically modified mice with increased endogenous n3 PUFA production had a smaller fat cell size compared with wild type mice [39]. Alongside these insights, endogenously synthesized n3 PUFAs could offer protective effects against abnormal fat storage and insulin resistance in mice that are fed high-fat diets [39,40]. We found that compared with the control group, the HMB treatment elevated the contents of C20:4n6 and C22:6n3, suggesting that HMB could not only reduce lipid accumulation, but also improve fatty acid composition in adipocytes.

Considering the beneficial effects of mitochondrial respiratory function on obesity, we next used the Seahorse Biosciences XF-24 Analyzer to determine the effects of HMB on the mitochondrial respiratory function of 3T3-L1 adipocytes. Basal respiration occurs in cells at rest for basic energy needs. In the presence of mitochondrial ATP synthase inhibitor oligomycin, HMB-treated cells showed a greater capacity to produce ATP. Moreover, the HMB treatment unlocked the potential of mitochondrial oxygen consumption. The enhancement of mitochondrial respiratory function was shown to increase FAO [41]. Our observations reveal that the adipocytes in the HMB group exhibited improved mitochondrial respiratory function, as evidenced by enhanced basal respiration, ATP production, and maximal respiration. These findings are in line with our earlier studies in C2C12 myotubes [42]. Interestingly, previous studies also show that HMB can promote the mitochondrial biogenesis of myocytes [42,43]. In addition, the improved mitochondrial respiratory function we report here in response to HMB treatments are perfectly in line with a previous study from Sun et al. [44]. In their study, Sun et al. [44] showed that leucine can promote mitochondrial biogenesis in the 3T3-L1 adipocytes. Given the overlap between leucine and HMB, it is tempting to infer that HMB has a similar function to leucine in regulating mitochondrial function. Mechanically, it was reported that PGC-1α is a crucial determinant of mitochondrial biogenesis, cellular respiration, and FAO [34,45]. Since HMB provoked a marked increase in PGC-1α expression, this might imply that HMB boosted the mitochondrial biogenesis and FAO of the 3T3-L1 adipocytes. Beyond that, UCP1 expression, a typical indicator of brown and beige adipocytes, is under the positive control of PGC-1α [23,46]. The activation of UCP1 can uncouple mitochondrial respiration from ATP synthesis and raise the rate of mitochondrial respiration, leading to increased heat production and energy expenditure [18]. More recent studies identified that the transcriptional basis of brown fat determination is mediated via the PRDM16/PGC-1α/UCP1 cascade [20,21,22]. Consistently, we observed parallel increased trends in PRDM16/PGC-1α/UCP1 expression in the HMB group, as the metabolites of leucine, KIC, and HMB share similar functions, such as promoting protein synthesis, preventing protein degradation, and regulating energy homeostasis [47]. Interestingly, KIC was demonstrated to promote fat browning in this study. However, this beneficial effect of KIC was inferior to that of HMB. Likewise, HMB was proven to be more efficacious in preventing protein degradation and potentiating mitochondrial respiration relative to KIC [42,48,49]. Hence, we recognize HMB as an agent that might be effective against muscle atrophy and obesity. Collectively, these results suggest that HMB augmented the mitochondrial function, mitochondrial biogenesis, and fat browning of the 3T3-L1 adipocytes. Additionally, mechanistically, the metabolic benefits may be correlated with the PRDM16/PGC-1α/UCP1 pathway.

Apart from the regulation of lipid metabolism, beige adipocytes also aid in the improvement of glucose homeostasis [50]. The benefits of fat browning were exemplified in germ-free mice receiving cold microbiota transplantation. These mice had increased insulin sensitivity and fat loss, and the beneficial effects were partly mediated by the activation of beige adipocytes [51]. In obese and insulin-resistant mice, the formation of beige adipocytes ameliorated glucose tolerance and reduced fasting insulin [52]. Similar observations were made in a recent study showing that beige adipocyte development prevented obesity and mitigated insulin resistance [53]. One major contributor to insulin resistance is the excess free fatty acids released by enlarged adipose tissue [54]. In this regard, we speculate that HMB could improve insulin sensitivity via the recruitment of beige adipocytes and the clearance of lipids.

By using 3T3-L1 cells, we examined the effects of HMB on lipid metabolism, mitochondrial function, and fat browning. Additionally, we discussed possible mechanisms and provided rational explanations. However, we might also call into question the generalization of these findings, because little information has been made available to offer direct support for our results obtained in the adipocyte cell culture setting. To the best of our knowledge, this is the first study to evaluate the effectiveness of HMB to regulate lipid metabolism in a 3T3-L1 cell model. Further studies are warranted to corroborate our findings and to define the role of HMB in obesity management.

## 5. Conclusions

In summary, the current study suggests that HMB treatment exerted salutary effects on lipid metabolism in 3T3-L1 adipocytes, such as inhibiting fat deposition, enhancing mitochondrial function, and accelerating fat browning. It also provides evidence that the activation of AMPK/Sirt1-mediated PPARγ signaling and PRDM16/PGC-1α/UCP1 pathways mediated the anti-obesity effect of HMB. Moreover, HMB may increase insulin sensitivity by improving fatty acid composition and may promote fat browning. HMB administration can be perceived as an effective strategy for treating obesity and insulin resistance, and our findings provide new insights into the development of novel drug therapies.

## Figures and Tables

**Figure 1 nutrients-15-02550-f001:**
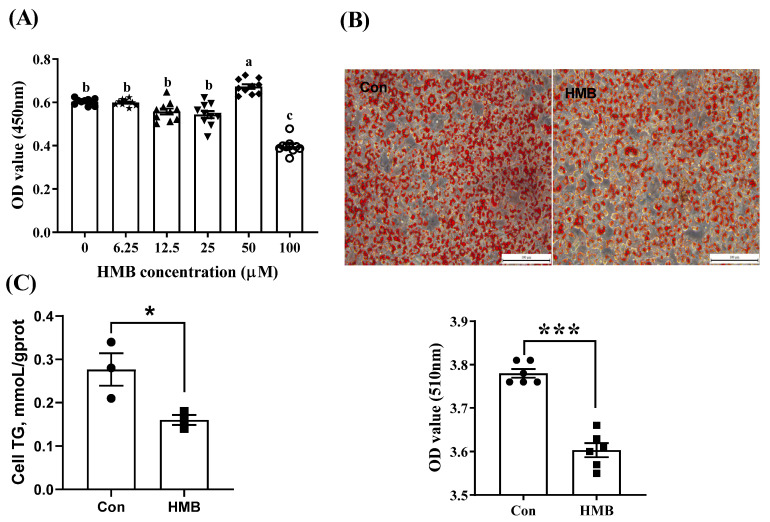
The effects of HMB treatment on the proliferation and lipid accumulation of 3T3-L1 cells. (**A**) Cell proliferation analysis (n = 10). Preadipocytes were treated with 0–100 µM of HMB for 24 h. (**B**) Oil red O staining. HMB (50 µM) was added to the differentiation medium for 48 h (n = 6). (**C**) Cellular TG levels. HMB (50 µM) was added to the differentiation medium for 48 h (n = 3). Solid square, circle, star, equilateral triangle, inverted triangle, diamond and hollow circle represent the data obtained by different HMB treatment respectively. Values are means, with their standard errors represented by vertical bars. ^a,b,c^ Values with different letters differed significantly (*p* < 0.05) by Duncan’s multiple comparisons. Significance levels were marked as * for *p* < 0.05 and *** for *p* < 0.001 via two-tailed *t*-test.

**Figure 2 nutrients-15-02550-f002:**
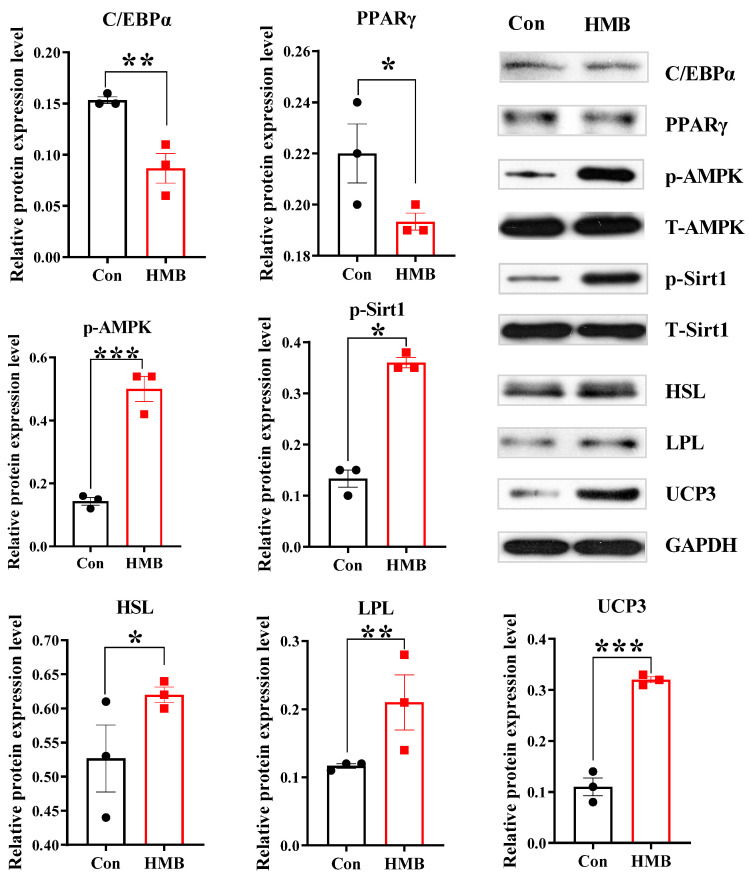
The effects of HMB on the expression of proteins relating to lipid metabolism in 3T3-L1 adipocytes (n = 3). Black circles and red squares represent the data obtained without and with HMB treatment, respectively. HMB (50 µM) was added to the differentiation medium for 48 h. T-AMPK level was used as a control for p-AMPK level.; T-Sirt1 level was used as a control for p-Sirt1; GAPDH level was used as a control for other protein levels. Values are means, with their standard errors represented by vertical bars. Significance levels were marked as * for *p* < 0.05, ** for *p* < 0.01, and *** for *p* < 0.001 via two-tailed *t*-test.

**Figure 3 nutrients-15-02550-f003:**
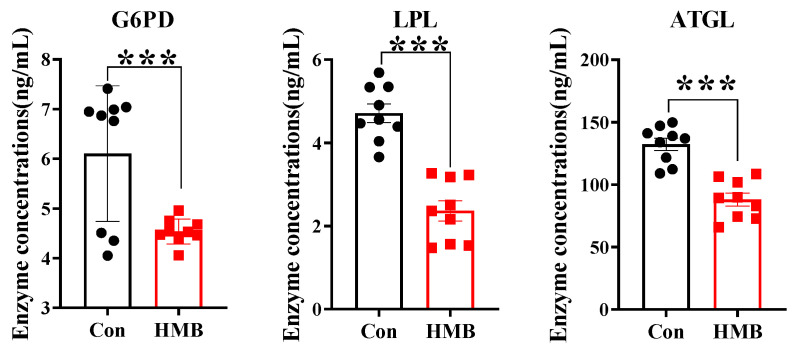
The effects of HMB treatment on lipid metabolism-related enzyme concentrations in 3T3-L1 adipocytes (n = 9). Black circles and red squares represent the data obtained without and with HMB treatment, respectively. Values are means, with their standard errors represented by vertical bars. Significance levels were marked as *** for *p* < 0.001 via two-tailed *t*-test.

**Figure 4 nutrients-15-02550-f004:**
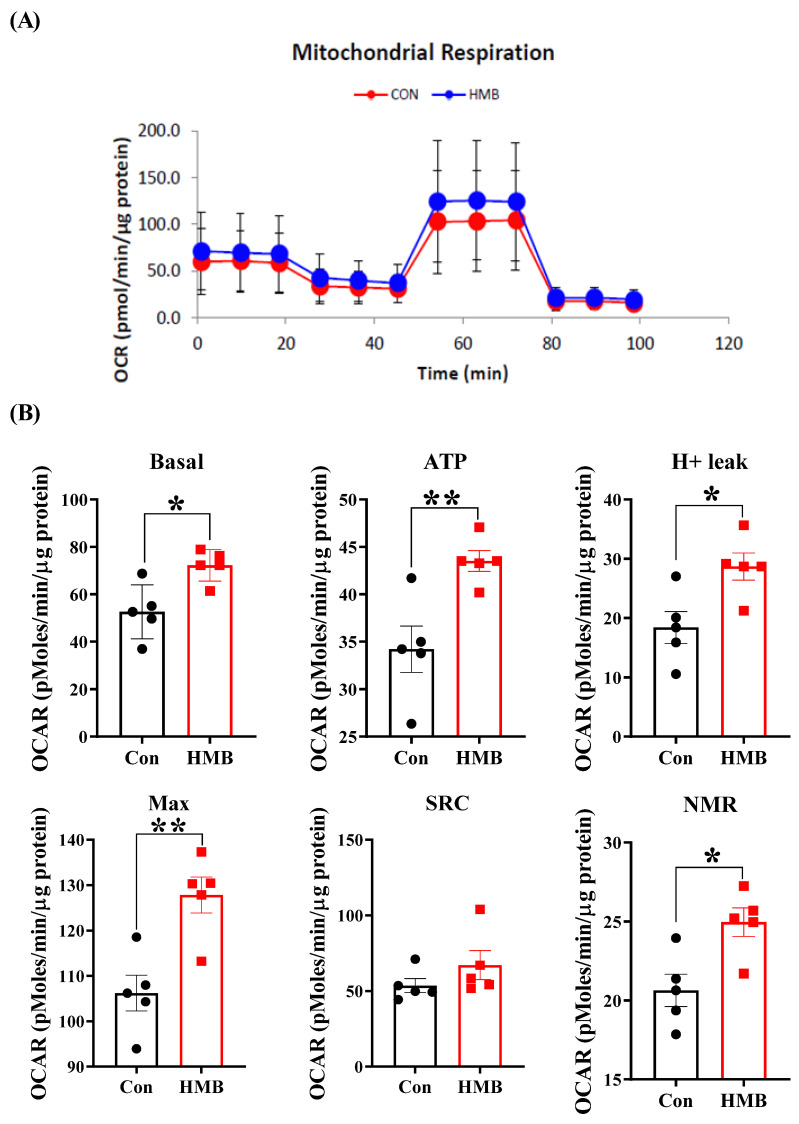
The effects of HMB treatment on the mitochondrial function of 3T3-L1 adipocytes (n = 5). (**A**) Mitochondrial OCR curves with or without HMB treatment. (**B**) Basal, basal respiration; ATP, ATP production; H^+^ leak, proton leak; Max, maximal respiration; SRC, spare respiration capacity; and NMR, non-mitochondrial respiration of 3T3-L1 adipocytes with or without HMB treatment, respectively. Black circles and red squares represent the data obtained without and with HMB treatment, respectively. When cells fully differentiated into adipocytes, they were treated with HMB (50 µM) for 24 h. Values are means, with their standard errors represented by vertical bars. Significance levels were marked as * for *p* < 0.05 and ** for *p* < 0.01 via two-tailed *t*-test.

**Figure 5 nutrients-15-02550-f005:**
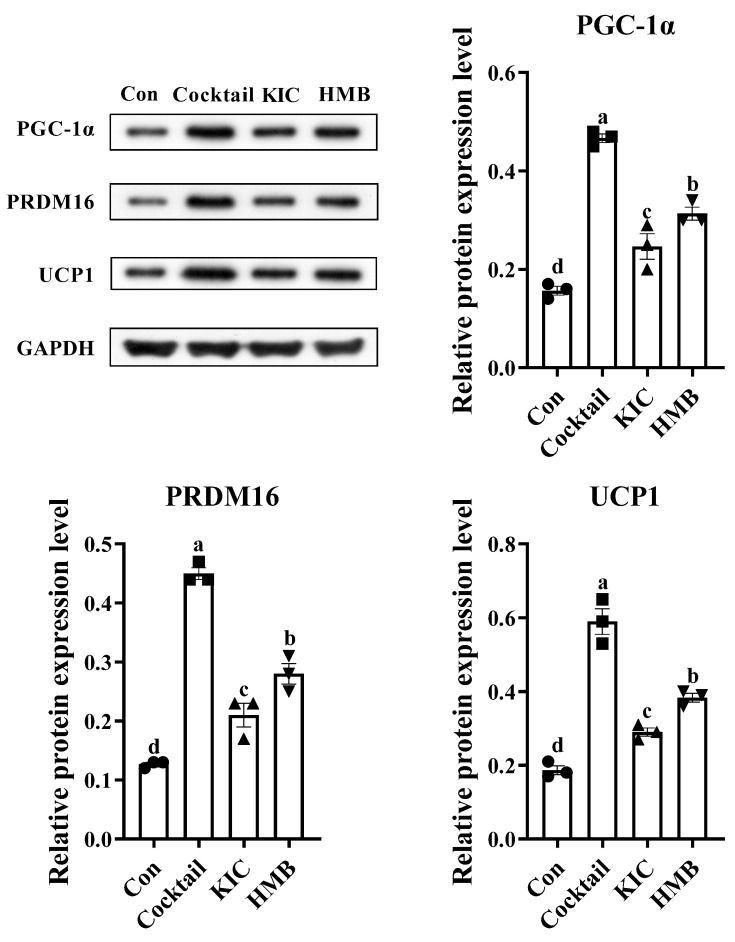
The effects of HMB on the expression of beige fat-specific markers (n = 3), where 3T3-L1 adipocytes were treated with browning cocktail (50 nM triiodothyronine and 1 µM rosiglitazone), KIC (0.5 mM), or HMB (50 µM) for 72 h. Solid square, circles, regular triangles, and inverted triangles represent the data obtained by different treatment respectively. Values are means, with their standard errors represented by vertical bars. ^a,b,c,d^ Values with different letters differed significantly (*p* < 0.05) by Duncan’s multiple comparisons.

**Table 1 nutrients-15-02550-t001:** The effects of HMB treatment on fatty acid composition (%) of 3T3-L1 adipocytes (n = 3).

Item	Control	HMB	SEM	*p*-Value
C10:0	0.61 ^b^	0.90 ^a^	0.00076	<0.05
C14:0	3.71 ^a^	2.68 ^b^	0.00254	<0.05
C15:0	8.12 ^a^	5.45 ^b^	0.00655	<0.05
C16:0	31.71	35.24	0.01101	0.11
C16:1	21.38 ^a^	12.60 ^b^	0.02117	<0.01
C17:0	2.41 ^a^	2.06 ^b^	0.00090	<0.05
C18:0	12.31 ^b^	21.28 ^a^	0.02119	<0.01
C18:1n9c	13.06	11.96	0.00739	0.55
C22:1n9	2.53	4.10	0.00502	0.12
C20:4n6	1.56 ^b^	2.15 ^a^	0.00132	<0.001
C22:6n3	0.68 ^b^	0.93 ^a^	0.00057	<0.001
∑n6 PUFAs	1.56 ^b^	2.15 ^a^	0.00132	<0.001
∑n3 PUFAs	0.68 ^b^	0.93 ^a^	0.00057	<0.001

^a,b^ Values within a row with different letters differed significantly by two-tailed *t*-test.

## Data Availability

The data presented in this study are available upon request from the corresponding authors.

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
