# Peer review of "β-Hydroxy-β-methyl Butyrate Regulates the Lipid Metabolism, Mitochondrial Function, and Fat Browning of Adipocytes"

_nutrients, 2023, doi:10.3390/nu15112550_

Round 1
Reviewer 1 Report (Previous Reviewer 3)
Upon reviewing the revised version of your manuscript, it is clear that some efforts have been made to improve the quality since its prior submission. However, the manuscript still falls short in certain aspects.
The introduction and discussion sections lack the necessary fluidity and cohesion, which detracts from the overall readability and comprehension. I strongly suggest these sections undergo thorough extra editing and revision to improve the logical flow of ideas.
Additionally, Figure 1a is still lacking the crucial bright field image. I understand that cells were utilized for the plate reader in this figure. However, this is not a valid reason to exclude such an important visual aid. It is imperative to repeat the experiment and record the morphological differences upon treatment with various HMB dosages. The addition of this visual data is non-negotiable for a comprehensive understanding of your findings.
None
Author Response
Dear Editors:
Thanks for the constructive suggestions and comments from Editorial Boarding and reviewers. We have read the referees' comments very carefully, have consulted and discussed the reviewers' comments with several professors, and now we have further revised the comments according to the reviewers' suggestions to improve the manuscript. At the same time, we indexed revisions in red color in the manuscript. Response to the comments of 2286210 was listed following with ‘A’ for answers.
Thank you very much for your considering our manuscript. We are looking forward to hearing from you soon.
Best regards.
Yours,
Yehui Duan
Reviewer #1
The introduction and discussion sections lack the necessary fluidity and cohesion, which detracts from the overall readability and comprehension. I strongly suggest these sections undergo thorough extra editing and revision to improve the logical flow of ideas.
A: Thanks for the reviewer’s suggestion. We have rewritten the introduction and discussion sections to improve the necessary fluidity and cohesion.
Additionally, Figure 1a is still lacking the crucial bright field image. I understand that cells were utilized for the plate reader in this figure. However, this is not a valid reason to exclude such an important visual aid. It is imperative to repeat the experiment and record the morphological differences upon treatment with various HMB dosages. The addition of this visual data is non-negotiable for a comprehensive understanding of your findings.
A: Thanks for the reviewer’s suggestion. As the reviewer said, the morphological differences after treatment with various HMB doses may actually assist in understanding our findings. It was a pity that Cell viability was assessed using cell-counting kit CCK-8 with an enzyme-linked immunosorbent assay plate reader (Bio-Tek, Winooski, VT, USA) in this experiment. The results were expressed as optical density (OD450) value without bright field cell images.
Reviewer 2 Report (Previous Reviewer 2)
All the images are not particularly clear, with low resolution. The horizontal and vertical axes are not clear, and there is overlapping of some significant symbols and data.
Author Response
Dear Editors:
Thanks for the constructive suggestions and comments from Editorial Boarding and reviewers. We have read the referees' comments very carefully, have consulted and discussed the reviewers' comments with several professors, and now we have further revised the comments according to the reviewers' suggestions to improve the manuscript. At the same time, we indexed revisions in red color in the manuscript. Response to the comments of 2286210 was listed following with ‘A’ for answers.
Thank you very much for your considering our manuscript. We are looking forward to hearing from you soon.
Best regards.
Yours,
Yehui Duan
Reviewer #2
All the images are not particularly clear, with low resolution. The horizontal and vertical axes are not clear, and there is overlapping of some significant symbols and data.
A: Thanks for the reviewer’s suggestion. We have improved all the figures.
Reviewer 3 Report (Previous Reviewer 1)
The author has improved most of the proposed revisions, but a few need to be improved. For example, different reagents may have different experimental results. In order to ensure the repeatability of the experiment, batch number should be provided, which is also an important reflection of whether the reagent is expired. Fat cells should provide units, not as explained by the authors; The absence of a positive control group should be analyzed in the discussion. It is suggested that it be modified and published.
Author Response
1Dear Editors:
Thanks for the constructive suggestions and comments from Editorial Boarding and reviewers. We have read the referees' comments very carefully, have consulted and discussed the reviewers' comments with several professors, and now we have further revised the comments according to the reviewers' suggestions to improve the manuscript. At the same time, we indexed revisions in red color in the manuscript. Response to the comments of 2286210 was listed following with ‘A’ for answers.
Thank you very much for your considering our manuscript. We are looking forward to hearing from you soon.
Best regards.
Yours,
Yehui Duan
Reviewer #3
The author has improved most of the proposed revisions, but a few need to be improved. For example, different reagents may have different experimental results. In order to ensure the repeatability of the experiment, batch number should be provided, which is also an important reflection of whether the reagent is expired.
A: Thanks for the reviewer’s suggestion. We have added batch numbers (Line 120, 127-143).
Fat cells should provide units, not as explained by the authors;
A: Thanks for the reviewer’s suggestion. We have provided units of fat cells in Line 109. If it is not what the reviewer required, could you kindly provide a more detailed description of your suggestion?
The absence of a positive control group should be analyzed in the discussion. It is suggested that it be modified and published.
A: Thanks for the reviewer’s suggestion. We have added the related contents in the discussion (Lines 267-271 and 310-315)

Reviewer 4 Report (New Reviewer)
In the manuscript titled “β-hydroxy-β-methyl butyrate regulates the lipid metabolism, mitochondrial function, and fat browning of adipocytes”, the authors Duan et al. show that 50uM β-hydroxy-β-methyl butyrate promotes proliferation of preadipocytes, reduced triglyceride accumulation, and improved mitochondrial respiration in differentiated adipocytes along with increased browning. The study is straightforward and most of the results are alright, but the authors need to make minor revisions to the manuscript especially in the methods section.
1. For the cell proliferation analysis, please mention the duration of starvation.
2. For the Oil Red O staining section, how was the absobance measured? Directly from stained cells? Or was the oil red O stain extracted from the cells and then absorbance was measured?
3. For the TG assay, were the cells lysed in any kind of lysis buffer?
4. Since G6PD, LPL and ATGL are not secreted proteins, I am assuming the ELISA was done using cell lysates? Please describe how cells were lysed and how the protein levels were normalized.
5. Please specify the temperature samples were jeated at in isooctane in method 2.7
6. Can the authors raise the position of the asterix denoting significance in their figures? Asterix that overlap with data points and error bars are harder to see.
7. The western blot images in Fig 2 are very low quality and are also too stretched. Can the authors show better westrn blot images?
8. The authors need to show total AMPK protein levels in Fig 2 as a control to show that pAMPK levels increased compared to total AMPK levels.
Author Response
Dear Editors:
Thanks for the constructive suggestions and comments from Editorial Boarding and reviewers. We have read the referees' comments very carefully, have consulted and discussed the reviewers' comments with several professors, and now we have further revised the comments according to the reviewers' suggestions to improve the manuscript. At the same time, we indexed revisions in red color in the manuscript. Response to the comments of 2286210 was listed following with ‘A’ for answers.
Thank you very much for your considering our manuscript. We are looking forward to hearing from you soon.
Best regards.
Yours,
Yehui Duan
Reviewer #4
- For the cell proliferation analysis, please mention the duration of starvation.
A: Thanks for the reviewer’s suggestion. We have mentioned the duration of starvation, which was 6 h. (Line 105)
- For the Oil Red O staining section, how was the absobance measured? Directly from stained cells? Or was the oil red O stain extracted from the cells and then absorbance was measured?
A: It was the oil red O stain extracted from the cells and then absorbance was measured.
- For the TG assay, were the cells lysed in any kind of lysis buffer?
A: For the TG assay, after collecting cell samples, 0.2 ~ 0.3 ml PBS (0.1 mol / L, pH7 ~ 7.4) was added to homogenize, and sonicated under ice water bath conditions (power: 300 W, 3 ~ 5 s/time, interval 30 s, repeat 3 ~ 5 times). Apart from cell disruption, lysis buffer (Triton X-100, 1-2%, 30-40 minutes) can also be used.
- Since G6PD, LPL and ATGL are not secreted proteins, I am assuming the ELISA was done using cell lysates? Please describe how cells were lysed and how the protein levels were normalized.
A: According to the manufacturer 's instructions (Jiangsu Meibiao Biotechnology Co., Ltd, Jiangsu, China), when detecting the intracellular components, the cell suspension was diluted with PBS (PH7.2-7.4), and the cell concentration reached about 1 million / ml. A portion of cells dissolved in PBS was used to detect the enzyme concentrations. Through repeated freezing and thawing, the cells are destroyed and the intracellular components are released. The remaining cells were used to detect the protein levels using the Enhanced BCA protein assay kit (P0009, Beyotime Biotechnology, China), which used for normalization.
- Please specify the temperature samples were jeated at in isooctane in method 2.7
A: Thanks for the reviewer’s suggestion. the temperature was 45℃.
- Can the authors raise the position of the asterix denoting significance in their figures? Asterix that overlap with data points and error bars are harder to see.
A: Thanks for the reviewer’s suggestion. We have improved all the figures.
- The western blot images in Fig 2 are very low quality and are also too stretched. Can the authors show better western blot images?
A: Thanks for the reviewer’s suggestion. We have improved all the western blot images.
- The authors need to show total AMPK protein levels in Fig 2 as a control to show that p-AMPK levels increased compared to total AMPK levels.
A: Thanks for the reviewer’s suggestion. We have shown the total AMPK protein levels in Fig 2 as a control to show that p-AMPK levels increased compared to total AMPK levels.
This manuscript is a resubmission of an earlier submission. The following is a list of the peer review reports and author responses from that submission.
Round 1
Reviewer 1 Report
This passage report that β-hydroxy-β-methyl butyrate regulates lipid metabolism, mitochondrial function and adipose Browning in 3T3-L1 cells. However, some parts of the paper need to be modified and perfected. For example, the abstract should be written in the format of purpose, method, result and conclusion.Compress the preface appropriately;The reagents used for research should have batch number or article number;3T3-L1 preadipocytes should provide units;The number of holes for parallel tests should be provided in the method;Why not have a positive control group?Sample numbers should be provided for all results, which is the basis of statistics;The references should be based on those of the last three years, and it is suggested to revise and perfect them for publication.
Author Response
- The abstract should be written in the format of purpose, method, result and conclusion. Compress the preface appropriately.
A: Thanks for the reviewer’s suggestion. We have revised the abstract according to the format of this journal.
- The reagents used for research should have batch number or article number.
A: Thanks for the reviewer’s suggestion. Based on our previous experience, the batch number may be not necessary. And we have provided enough details. So the information about our reagents meets the requirement of this journal.
- 3T3-L1 preadipocytes should provide units.
A: Thanks for the reviewer’s suggestion. We have added “3T3-L1 preadipocytes were seeded in a 96-well plate (2000 cells per well).”
- The number of holes for parallel tests should be provided in the method.
A: Thanks for the reviewer’s suggestion. We have added sample numbers in the captions of figures and tables.
- Why not have a positive control group?
A: We failed to consider this factor when we designed this study. Thanks for the reviewer’s suggestion. We would like to have a positive control group in our future studies.
- Sample numbers should be provided for all results, which is the basis of statistics.
A: We have added sample numbers in figures and tables.
- The references should be based on those of the last three years, and it is suggested to revise and perfect them for publication.
A: Thanks for the reviewer’s suggestion. Although some references have been published for years, they are fundamental research and are still a classical proof of relevant topics. They are in support of what we want to convey. So, we can retain them.

Reviewer 2 Report
The author tried to present the role of β-hydroxy-β-methyl butyrate on lipid metabolism, mitochondrial function, and fat browning of 3T3-L1 cells. However, the results can not support solid conclusions, and more experiments are required. Furthermore, the manuscript should be subjected to thorough English Editing before the next submission.
I have the following minor recommendations:
1. Based on the provided statement, it appears that a suitable HMB concentration was tested in 3T3-L1 preadipocytes, but it is not clear if this concentration is also suitable for 3T3-L1 adipocytes. Is this concentration also suitable for 3T3-L1 adipocytes? In addition, Why high concentration HMB inhibited cell proliferation?
2. The effect of HMB on lipid metabolism was done in 3T3-L1 preadipocytes, but the role of HMB on mitochondrial respiratory function was determined in 3T3-L1 adipocytes. Preadipocytes and adipocytes have great differences. The effect of HMB on lipid metabolism or mitochondrial respiratory might be different between preadipocytes and adipocytes. I think additional experiments are required.
3. Figures are not very clear, especially 2.
4. For lipid accumulation, in addition to oil red O staining, did you test TC or TG content in the cell or medium?
5. Did you test cell viability after HMB treatments?
6. How many times was the experiment done?
7. Does HMB affect the differentiation of this cell?
8. Numerous studies have shown that adipocytokines play a key role in adipocytes. Does HMB affect the secretion of adipocytokines?
Reviewer 3 Report
The authors of this paper have assessed the function of HMB in adipogenesis using the in vitro model. The browning effect of HMB in adipocyte might be informative for the field. However, several issues in this paper need to be addressed before being considered for publication.
I didn't follow the rationale why author choose to perform in vitro model to access the function of HMB in adipogenesis when in vivo model is feasible. It's not clear to me whether the purpose it's to avoid potential non-tissue-autonomous effect or not? If not, why not consider performing HMB treatment in vivo, evaluate the browning effect of HMB on both inguinal and visceral WAT? Authors need to explain their rationale for experiment design better.
Many of figures they shown in this manuscript needs some improvement so reviewer can interpret their results better.
1) Bright field cell images are required for Fig1A.
2) Please properly label all Western Blot image in Figure 2 and Figure4- specify treatment group on top panel of western blot.
3) Specify treatment group on top panel of Fig1B
4) All labeling in Fig2 is too small to be read, including the significant difference mark.
5) It's not clear to me what comparison was performed in Figure 4 for qPCR analysis. Whether each group was compared with control group or different analysis was performed. The sentence-"a,b,c,dValues with different letters differed significantly (P < 0.05) by Duncan's multiple comparisons."-doesn't make sense.
Minor issues:
Authors may consider changing all bar graphs used in this manuscript to show dots for individual samples.
The logic of this paper is a bit hard to follow, in part because of the experiment result interpretation, but also because the writing needs a lot of improvement.
Reviewer 4 Report
The authors should format the paper following Journal guidelines including references, figures and Tables.
The novelty character of paper should be marked.
A graphical scheme of study approach should be inserted.
Some introductory lines should be added in Section results to better introduce the different type of results
Results in Figure 2 should be better described in the text.
Conclusion should be implemented: considerations, limits, advantages and future directions should be marked.
Author Response
Please see the figures in attachment.
- The authors should format the paper following Journal guidelines including references, figures and Tables.
A: Thanks for the reviewer’s suggestion. We have formatted the paper following Journal guidelines.
- The novelty character of paper should be marked.
A: We talk about this part in “Abstract” and “Discussion”. As far as we know, this is the first report showing the effectiveness of HMB to regulate energy metabolism in a 3T3-L1 cell model. In addition, we discussed possible mechanisms and provided rational explanations.
- A graphical scheme of study approach should be inserted.
A: Thanks for the reviewer’s suggestion. We have made sample schemes.
- Some introductory lines should be added in Section results to better introduce the different type of results.
A:. Thanks for the reviewer’s suggestion. We have added related information.
- Results in Figure 2 should be better described in the text.
A: Thanks for the reviewer’s suggestion. We have added related information. “These differences might indicate blunted lipogenic reaction and augmented catabolism of lipids, which caused reduced lipid accumulation in 3T3-L1 preadipocytes.”
- Conclusion should be implemented: considerations, limits, advantages and future directions should be marked.
A: We wrote these contents in “Discussion”: “By using 3T3-L1 cells, we examined the effects of HMB on lipid metabolism, mitochondrial function, and fat browning. Additionally, we discussed possible mechanisms and provided rational explanations. But it might also call into question the generalization of these findings, because little information has been made available to offer direct support for our results obtained in the adipocyte cell culture setting. To the best of our knowledge, this is the first study to evaluate the effectiveness of HMB to regulate energy metabolism in a 3T3-L1 cell model. Further studies are warranted to corroborate our findings and define the role of HMB in obesity management.”

Round 2
Reviewer 2 Report
I do not find the author's response to be satisfactory in addressing my concerns.
Author Response
- Based on the provided statement, it appears that a suitable HMB concentration was tested in 3T3-L1 preadipocytes, but it is not clear if this concentration is also suitable for 3T3-L1 adipocytes. Is this concentration also suitable for 3T3-L1 adipocytes? In addition, Why high concentration HMB inhibited cell proliferation?
A: Thanks for the reviewer’s suggestion. 1) Based on the cell viability and oil O red experiments, we found that 50 µM of HMB could increase the proliferation of 3T3-L1 preadipocytes and inhibit the differentiation of 3T3-L1 adipocytes, thus we regarded it as a suitable HMB concentration. 2) We have no idea why high concentration HMB inhibited cell proliferation, one possible explanation was that a high dose of HMB was toxic to cells.
- The effect of HMB on lipid metabolism was done in 3T3-L1 preadipocytes, but the role of HMB on mitochondrial respiratory function was determined in 3T3-L1 adipocytes. Preadipocytes and adipocytes have great differences. The effect of HMB on lipid metabolism or mitochondrial respiratory might be different between preadipocytes and adipocytes. I think additional experiments are required.
A: Thanks for the reviewer’s suggestion. In fact, we investigated the roles of HMB on mitochondrial respiratory function in both 3T3-L1 preadipocytes and adipocytes. The related results using 3T3-L1 preadipocytes were added and shown in Figure 3.
- Figures are not very clear, especially 2.
A: Thanks for the reviewer’s suggestion. We have improved figures.
- For lipid accumulation, in addition to oil red O staining, did you test TC or TG content in the cell or medium?
A: TG content in cells were analyzed and shown in Figure 1(c).
- Did you test cell viability after HMB treatments?
A: Yes, we tested cell viability after HMB treatments, and the results were shown in Figure 1(A).
- How many times was the experiment done?
A: Three.
- Does HMB affect the differentiation of this cell?
A: Yes. The aim of determining changes (Oil Red O staining and WB) in the preadipocytes was to explore whether HMB could inhibit lipid accumulation and adipogenesis during differentiation. The results in Figure 1(B, C) and Figure 2 suggested that HMB inhibited lipid accumulation during differentiation.
- Numerous studies have shown that adipocytokines play a key role in adipocytes. Does HMB affect the secretion of adipocytokines?
A: We have noticed this point. Our previous study showed that HMB upregulated serum adiponectin concentrations in growing pigs. “β-Hydroxy-β-methylbutyrate modulates lipid metabolism in adipose tissues of growing pigs”

Reviewer 3 Report
I don't think the authors addressed my comments very well in this case. In the author's response letter, they mentioned, 'We did not attempt to avoid potential non-tissue-autonomous effects. In fact, we have previously conducted experiments on mice. HMB administration reversed the HFD-induced whitening and enhanced the protein expression of thermogenic marker UCP1 in BAT. "Gut microbiota mediates the protective effects of dietary β-hydroxy-β-methylbutyrate (HMB) against obesity induced by high-fat diets." Because there is no available information about HMB effects in 3T3-L1 cells, we performed the present study.' The author missed the point about the clarification of the rationale for their experimental design.
If a previous study has already well-characterized the browning effect of HMB in white adipocytes in vivo, the novelty of this study will be significantly decreased. The major benefit of doing in vitro adipogenesis assays is that they can avoid non-tissue-autonomous regulation - that is, whole-body metabolic effects outside of adipose tissue - which confirms any potential drug effect in targeting specific cell types (white adipocytes in this study). The previous study referred to didn't assess the browning effect of HMB in white adipocytes; instead, it focused on brown adipocytes and showed that HMB can prevent lipid accumulation after high-fat diet treatment in vivo. In this case, the rationale for doing in vitro analysis would be different.
In my opinion, the authors didn't address these possibilities well, nor did they introduce their rationale for the experimental design clearly. Moreover, my other comments were not adequately addressed by the authors. The labeling in Fig2 is still too small to be seen clearly, including the significance markers. All labeling in figures should be consistent in font. I think those are really basic requirements to make scientific figures look professional.
Regarding minor issues, showing individual dots on bar graphs does not conflict with showing the mean value. Showing individual dots on bar graphs helps readers see the individual differences.
I couldn't see significant improvement in the scientific writing part. Authors need to learn to better interpret their results and provide clear conclusions in English so readers can understand what they want to show.
Author Response
- I didn't follow the rationale why author choose to perform in vitro model to access the function of HMB in adipogenesis when in vivo model is feasible. It's not clear to me whether the purpose it's to avoid potential non-tissue-autonomous effect or not? If not, why not consider performing HMB treatment in vivo, evaluate the browning effect of HMB on both inguinal and visceral WAT? Authors need to explain their rationale for experiment design better.
A: In this study, in vitro models were used to avoid potential non-tissue-autonomous effects. Because there is no available information about HMB effects on the 3T3 L1 cell model, we designed and conducted this study.
- Bright field cell images are required for Fig1A.
A: Cell viability was assessed using cell-counting kit CCK-8 with an enzyme-linked immunosorbent assay plate reader (Bio-Tek, Winooski, VT, USA). The results were expressed as optical density (OD450) value without bright field cell images.
- Please properly label all Western Blot image in Figure 2 and Figure4- specify treatment group on top panel of western blot.
A: Thanks for the reviewer’s suggestion. We have improved all figures.
- Authors may consider changing all bar graphs used in this manuscript to show dots for individual samples.
A: Thanks for the reviewer’s suggestion. We have changed the format.
- Authors need to learn to better interpret their results and provide clear conclusions in English so readers can understand what they want to show.
A: Thanks for the reviewer’s suggestion. We have revised the manuscript.
